# Predicting Carcass Weight of Grass-Fed Beef Cattle before Slaughter Using Statistical Modelling

**DOI:** 10.3390/ani13121968

**Published:** 2023-06-12

**Authors:** Kalpani Ishara Duwalage, Moe Thandar Wynn, Kerrie Mengersen, Dale Nyholt, Dimitri Perrin, Paul Frederic Robert

**Affiliations:** 1Centre for Data Science, Queensland University of Technology, Brisbane 4000, Australia; m.wynn@qut.edu.au (M.T.W.); k.mengersen@qut.edu.au (K.M.); d.nyholt@qut.edu.au (D.N.); dimitri.perrin@qut.edu.au (D.P.); 2Black Box Co., Millstream, Brisbane 4888, Australia; fred@blackboxco.com.au

**Keywords:** grass-fed beef cattle, carcass weight, body weight, prediction, weaning, regression

## Abstract

**Simple Summary:**

The beef industry plays a crucial role in the livestock supply chain, and data are becoming increasingly vital for informed decision-making. In Australia, significant amounts of data are collected within cattle farms; however, due to a lack of suitable data-driven methods, much of the data go to waste without being effectively utilized. This study developed a statistical model to predict the carcass weight (CW) of grass-fed beef cattle at four different stages before slaughter using farm-level data. Two statistical modelling approaches were used, and results were compared. Four timespans prior to the slaughter, i.e., 1 month, 3 months, 9–10 months, and at weaning, were considered in the predictive modelling. Seven phenotypic features of cattle were used to describe the CW. The results showed that the CW of the cattle was strongly associated with the animal’s body weight at each stage before slaughter. The CW can be predicted with an average error of 4% (~12–16 kg) at three months before slaughter. The predictive error increased gradually when moving away from the slaughter date, where the prediction error at weaning was ~8% (~20–25 kg). The outcomes of this study demonstrate the value of using historical data in optimizing production and improving efficiency in the supply chain.

**Abstract:**

Gaining insights into the utilization of farm-level data for decision-making within the beef industry is vital for improving production and profitability. In this study, we present a statistical model to predict the carcass weight (CW) of grass-fed beef cattle at different stages before slaughter using historical cattle data. Models were developed using two approaches: boosted regression trees and multiple linear regression. A sample of 2995 grass-fed beef cattle from 3 major properties in Northern Australia was used in the modeling. Four timespans prior to the slaughter, i.e., 1 month, 3 months, 9–10 months, and at weaning, were considered in the predictive modelling. Seven predictors, i.e., weaning weight, weight gain since weaning to each stage before slaughter, time since weaning to each stage before slaughter, breed, sex, weaning season (wet and dry), and property, were used as the potential predictors of the CW. To assess the predictive performance in each scenario, a test set which was not used to train the models was utilized. The results showed that the CW of the cattle was strongly associated with the animal’s body weight at each stage before slaughter. The results showed that the CW can be predicted with a mean absolute percentage error (MAPE) of 4% (~12–16 kg) at three months before slaughter. The predictive error increased gradually when moving away from the slaughter date, e.g., the prediction error at weaning was ~8% (~20–25 kg). The overall predictive performances of the two statistical approaches was approximately similar, and neither of the models substantially outperformed each other. Predicting the CW in advance of slaughter may allow farmers to adequately prepare for forthcoming needs at the farm level, such as changing husbandry practices, control inventory, and estimate price return, thus allowing them to maximize the profitability of the industry.

## 1. Introduction

The Australian beef industry contributes significantly to the Gross Domestic Product [1]. In 2018, Australia accounted for approximately 4% of global beef production, and in 2019, Australia was the second largest beef exporter, after Brazil [2]. Approximately 64% of the total beef production in Australia in 2019 consisted of grass-fed cattle [3]. Moreover, grass-fed beef constitutes approximately 72% of exported beef [3]. Strategies to enhance the profitability and sustainability of grazing beef production are therefore important for the Australian economy.

In recent years, there has been a growing trend in the agricultural sector to utilize historical data for decision-making [4,5,6,7]. However, in Australia’s cattle farming industry, despite the collection of large volumes of data, its potential value remains largely untapped due to a lack of suitable analysis tools and data-driven insights. Moreover, many farmers in the grazing beef industry heavily rely on intuitive decision-making based on personal experiences and historical patterns, often without thorough analysis or consideration of factual information. In this context, harnessing historical data to develop data-driven tools presents significant opportunities for enhancing the efficiency of beef production.

To date, there has been a scarcity of published studies focusing on the prediction of the carcass weight (CW) in beef cattle [4,8,9,10]. Only one Australian research was found among these studies that discuss CW predictions [10]. However, this particular study predicted CW relative to the pre_slaughter weight and the dressing percentage (DP); the DP is calculated based on the hot carcass weight and the pre_slaughter weight [11]. While McPhee’s approach [10] can be influenced significantly by the cattle’s breed and other external factors, the advantages of the proposed method are also minimal due to its reliance on the pre_slaughter weight. The factors considered in the studies outside Australia include zoometric measurements of cattle [8], economic and nutritional factors, and environmental factors [4].

In this paper, we present a methodology to predict the CW of grass-fed beef cattle at four different stages before slaughter by using a set of phenotypic characteristics of cattle that are readily available at the farm level. We further discuss the impact of the weaning weight (WW) on the CW. In this study, we used two different statistical methods to develop and compare the predictions of the CW. The two models included a more traditional regression—the multiple linear regression model (MLR)—and the relatively novel and popular boosted regression tree model (BRT) [12,13].

The CW predictions provided by this study offer numerous advantages for grazing beef producers in Australia. Early forecasts enable adjustments in husbandry practices to enhance cattle growth. They also help in identifying the most suitable market for the cattle. Moreover, CW forecasts are valuable for predicting cash flow and ensuring timely fulfillment of customer orders. Additionally, these forecasts may contribute to minimizing unnecessary paddock costs and optimizing the overall operational efficiency.

## 2. Materials and Methods

### 2.1. Data

The study conducted in this research utilized data provided by Black Box Co. (Brisbane, Australia), which boasts the largest collection of phenotypic records for cattle across Australia’s supply chain. Notably, the northern beef industry, which accounts for more than half of Australian beef production, is a key focus. The dataset utilized in the study consisted of 2995 grass-fed beef cattle, which were slaughtered between 2019 and 2021. These cattle were sourced from three major cattle farms in the northern region of Australia in which various body weights at different stages from weaning to slaughter were recorded. The northern region of Australia is characterized by tropical and sub-tropical climates [14]. Consequently, the predominant breed in the northern beef industry is *Bos indicus* and its derivatives, which exhibit adaptability to these tropical conditions [15]. A comprehensive overview of the dataset is provided in the results section of the study.

### 2.2. Modelling and Forecasting

The outcome of interest (response variable) of the study was the CW. Weaning weight (WW), weight gain since weaning to each stage before slaughter (GSW), time (days) since weaning to each stage before slaughter (TSW), breed, sex, weaning season (dry and wet), and property were considered as potential explanatory variables (predictors) of the CW. Here, the sum of WW and GSW represented the BW of the cattle in the corresponding stage before slaughter. Weaning weight is the BW of the cattle at weaning and typically ranges between 100 kg and 350 kg [16]. TSW was considered a predictor alternative to the age of the cattle, where age was not utilized due to the higher degree of uncertainty associated with the date of birth records. The breed was encoded as a factor variable representing five major breeds: Brahman, Brahman cross, British/European cross, crossbreed/composite, Santa cross, and a category for unknown breeds; initially, seventeen different breeds were in the data, and they were reduced into these six types based on the desires of end users. The weaning season was encoded as a factor variable representing wet and dry seasons, where May to October and November to April were considered as dry and wet seasons, respectively.

The dataset was split into subsets based on the availability of BW data and the desires of end users to evaluate the predictive performance of the models at four different stages before slaughter. These four stages were chosen for the following reasons:At weaning: Since farmers often acquire cattle from different breeders at weaning, forecasting at this stage can provide an initial estimate of final production and assist in decision-making.Nine–ten months before slaughter: In northern cattle farming, the average time cattle spent in pastures from weaning to slaughter is around two years. Forecasting at this intermediate stage allows farmers to adjust husbandry practices if necessary to improve weight gain.Three months before slaughter: Forecasts at this stage are highly valuable for decision-making in the selling process, such as identifying an appropriate market for cattle and ensuring timely fulfillment of customer orders.One month before slaughter: These forecasts can support optimization of freight transport, enable predictions related to cash flow management, etc.

The forecasts in these distinct stages provide useful information for farmers, assisting them in making informed decisions at different points in the cattle production and selling cycle. Consequently, two subsets from the entire sample, containing BW recorded before 1 month of slaughter (set 1) and 9–10 months of slaughter (set 2), were considered and used to evaluate forecast performance before 1 month and 9–10 months of slaughter, respectively. The entire dataset (set 3) was considered to obtain the forecasts before 3 months of slaughter. Set 3 was further utilized to evaluate forecast performance at weaning.

In set 3, the majority of cattle had no BW recorded before 3 months of the slaughter, and the following procedure was used to approximate the BW:

BW before 3 months of slaughter =

^1^ Pre_slaughter BW − (91 − (slaughter date − pre_slaughter BW recorded date)) × ^2^ ADG

^1^ All the cattle had a pre_slaughter BW which was generally recorded within one to two weeks before the slaughter. ^2^ ADG is the average daily gain of the cattle and described as: ADG = (pre_slaughter BW − the nearest BW before pre_slaughter BW)/number of days between the two BW recorded dates.

For 1 month, 3 months, and 9–10 months before slaughter, the MLR and BRT models were fitted using all predictors: GSW, TSW, WW, sex, breed, weaning season, and property. To assess the predictive performance at weaning, models were fitted using five predictors only; GSW or TSW do not exist at weaning. In each modelling scenario, the predictive accuracies of the models were validated using separate test sets that were not used to train the models.

In addition to forecasting, we conducted a separate analysis for the entire dataset to explore how grass and labor costs varied during the post-weaning-to-slaughter period based on WW, weight gain, and time on grass; grass and labor are the main costs in grazing beef production in Northern Australia. However, the purpose of the present cost analysis was to compare the variations in main costs in the grazing beef industry relative to WW, weight gain, and time on grass, but not to provide estimates of associated costs.

### 2.3. Measures of Predictive Performance

The study used the mean absolute percentage error (MAPE), the mean absolute error (MAE), and the root mean squared error (RMSE) for evaluating the predictive accuracy of the models. The MAE computes the average absolute difference between the actual CW and the predicted CW, whereas the MAPE computes the absolute difference between the actual and predicted weights as a percentage respective to the actual CW. The RMSE computes the square root of the mean squared differences between the actual CW and the predicted CW. The equations for MAE, MAPE, and RMSE are as follows:$$MAE=\frac{1}{n}\;\sum_{i=1}^{n}\left|y_{i}-{\hat{y}}_{i}\right|,\;MAPE=\frac{1}{n}\;\sum_{i=1}^{n}\left|\frac{y_{i}-{\hat{y}}_{i}}{y_{i}}\right|,\;RMSE=\sqrt{\frac{\sum_{i=1}^{n}{\left(y_{i}-{\hat{y}}_{i}\right)}^{2}}{n}}$$
where $y_{i}$ is the observed CW and ${\hat{y}}_{i}$ is the predicted CW in the validation (test) dataset.

### 2.4. Multiple Linear Regression

Multiple linear regression [17] models the linear dependence between a response variable Y and a set of predictor variables X. The standard form of a multiple linear regression model is defined as follows:$$Y_{i}=\beta_{0}+\beta_{1}X_{i1}+\cdots +\;\beta_{k}X_{ik}+\varepsilon_{i},\;\;\;\;\;\;i=1,\ldots ,n$$
where $Y_{i}$ is the response variable, $X_{i}$ are the predictor variables, $\beta_{0}$ is the constant term, $\beta_{i}$ are the regression coefficients, and $\varepsilon_{i}$ is the residual error.

### 2.5. Boosted Regression Trees

Boosted regression trees (BRTs) are an emerging machine-learning technique utilized in modelling and forecasting. It combines two powerful techniques, namely regression trees and boosting, to enhance predictive outcomes [18]. Regression trees model the dependence between the response variable Y and a set of predictor variables X through a set of splits of the predictor variables X in a tree-like structure [18]. Boosting, on the other hand, combines multiple weak learners to create a stronger model. Consequently, the BRT model combines a weighted set of single trees for improved prediction performance [18]. Further, the BRT model effectively handles complex non-linear predictor–response associations, missing data and outliers [13,19]. The general form of a BRT can be written as follows [13,20]:$$f\left(X\right)=\sum_{p}\beta_{p}b\left(X;\;\gamma_{p}\right),\;p=1,2,\ldots \ldots ,P$$
where $b\left(X;\;\gamma_{p}\right)$ represent individual regression trees with $\gamma_{p}$ indicating the split variables and their values at splitting and terminal nodes, and $\beta_{p}$ represent the boosting mechanism through expansion coefficients or weight values assigned to the nodes of the regression trees.

A key step in BRT modelling is the regularization of the model parameters to reduce model overfitting. These parameters are the number of trees, tree complexity (the number of splits), and the learning rate (the contribution of each tree to the growing model). In this study, a series of BRT models were fitted with different combinations of the number of trees (200 to 2000), tree complexity (1 to 5 splits), and the learning rate (0.01 to 0.001).

The study used R statistical software to conduct all the analyses [21]. The package dismo [22] and R stats package were used for the BRT and the MLR models, respectively.

## 3. Results

### 3.1. Characteristics of Grass-Fed Beef Cattle

The majority of the beef cattle were crossbreeds, with Brahman crossbreeds comprising almost half (~49%) of the sample (Table 1). A quarter of cattle in the sample had no breed recorded, where a majority of them also were Brahman crossbreeds. The CW of the beef cattle slaughtered between the years 2019 and 2021 was 295 kg on average, varying from 197 kg to 426 kg (Figure 1A). The WW of the cattle was 217 kg on average (Figure 1B) with 10% early weaners (weaned at 100–150 kg). In the northern beef industry, weaning generally occurred on average (sd = 6 months) two years before slaughter. The cattle were weaned predominantly (~62%) in the dry season. They gained a mean weight of 336 kg between weaning and slaughter, with on average 0.5 kg per day. The average BW of the cattle at pre_slaughter was 554 kg. Only 2.7% of the beef cattle were female.

### 3.2. Analyses

The primary focus of interest in the study was forecasting performance in the four scenarios. However, the relative importance of predictors in forecasting the CW was briefly discussed first. The outcomes of the MLR and the BRT models across four modelling scenarios indicated that WW, GSW, TSW, and breed were significantly (*p* < 0.05) associated with the CW (Figure 2); in scenario 4 at weaning, the significant predictors were only WW (BW) and breed. The most variation in the CW was explained by the body weight (WW + GSW) of the cattle (Table 2 and Figure 2). BW alone explained above 70% of the variation in the CW within three months before slaughter, while it was 30–40% within 9–10 months before slaughter. BW at weaning (WW) also explained 8–10% of the variation in CW.

Sex was found to be a significant predictor of the CW through the modelling of the entire dataset (set 3); however, sex was excluded from the forecasting models due to the small proportion of females in the sample and the absence of female cattle in the smaller subsets (set 1 and set 2). The season of weaning and property showed no significant effect on the CW. Further, no significant interactions between predictors were found.

### 3.3. Predictive Performance

The error estimates (MAPE (%), MAE (kg), and RMSE (kg)) of the predictive models under four scenarios are reported in Table 3. Figure 3 graphically illustrates the forecasting performance using observed and predicted values with a 95% prediction interval.

Overall, the outcomes with the two methods, the MLR and the BRT, were approximately similar. The predictive ability of the CW decreased when moving away from the slaughter. Both models predicted the CW with an approximately 3% average error (~8–12 kg) one month prior to slaughter, while the MAPE was 4% (~12–16 kg) when predicting three months prior to slaughter. The error increased considerably (MAPE—~6% and ~16–21 kg) for CW predictions at 9–10 months before slaughter. At weaning, the CW could be predicted with an average error of ~8% (~20–25 kg) using BW and breed only.

Additional cost analysis (descriptive analysis) indicated that cattle with lower WWs were associated with higher growing costs (grass + labor) during the post-weaning-to-slaughter period due to a longer time spent on grass (Table 4). The costs were particularly higher for cattle weaned at 100–150 kg (early weaners). The association between early weaners and the CW was further reflected in modelling in scenario 4 (Figure 4), where cattle weaned at a lower BW were associated with lower CW. For instance, cattle weaned at 100 kg and cattle weaned at 200 kg had an approximately 35 kg difference in CW. However, the impact on the CW decreased with increasing weaning weights, e.g., the difference in CW among cattle weaned between 250 kg and 350 kg was less than 10 kg. However, the findings also indicate that the final CW and the associated costs can be impacted by weight gain during the post-weaning-to-slaughter period for some cows, regardless of the WW.

## 4. Discussion

The aim of the present study was to develop a statistical model to predict the CW of grass-fed beef cattle at different stages before slaughter by using data collected at the farm level. The key predictor of the CW in the models was the BW of the animal in the corresponding stage before slaughter. The results of the study revealed the potential for predicting the CW of grazing beef cattle with higher accuracy several months prior to slaughter. Furthermore, the findings demonstrated a good level of accuracy in predicting the CW even at the weaning stage.

The forecasting model developed in this study is beneficial for grazing beef producers in Australia in various ways; for instance, early forecasts may be useful in changing husbandry practices if needed to improve the growth of the cattle, such as the provision of supplementation. Further, these forecasts can be used to identify the appropriate market for cattle; for example, if unlikely to yield a high CW, those cattle may be sold into a feedlot or to live export. Forecasts of the CW may be also useful for predicting cash flow, identifying when management practices should take place, e.g., ensuring withholding periods on drugs are appropriate and optimizing freight transport such as grouping the appropriate number of cattle on a truck. Moreover, fulfilling customer orders in a timely manner and mitigating unnecessary paddock costs can be considered as further benefits of early forecasts of the CW. It is worth noting that forecasts prior to one month are less beneficial across the four scenarios examined in the study.

To the best of our knowledge, the present statistical methodology to predict the CW is a novel application to the northern Australian beef industry. The approach used in this study to predict the CW is applicable not only to Australia, but also to beef production systems in other tropical regions worldwide that record essential cattle data. Given that over half of global beef production occurs in tropical regions, with *Bos indicus* and its derivatives being the predominant breeds, the implications of these findings extend to a significant portion of the global beef industry [23].

In the literature, only a limited number of studies have specifically focused on predicting the CW of beef cattle. A study conducted in Spain [8] utilized zoometric measurements of beef cattle to predict the CW before six different stages of slaughter starting from 30 days to 150 days. The predictive errors (MAPE) of their models for one month and three months before slaughter were 3.3% and 3.9%, respectively, whereas the corresponding figures for our study were 2.7% and 4.0%, respectively; however, obtaining different zoometric measurements of animals is a difficult task compared to weighing the cattle. Hence, the approach proposed in our study is more straightforward, because collecting BW data of cattle is becoming more common on many farms. For instance, innovations in technology, such as walk-over weighing and Optiweigh methods are used in Australia currently. A recent study [4] conducted in Brazil used regression (multiple linear regression) and machine learning methods (random forest and multilayer neural networks) to predict the CW. This study incorporated a wide range of factors, including environmental conditions, economic considerations, nutritional aspects, and other relevant variables to predict the CW. However, the forecast error reported in the Brazil study was comparably larger and varied between 30 and 50 kg; instead, the prediction error of our models, which uses a few readily available factors, did not exceed 21 kg even with the forecasts before 9–10 months of slaughter.

An interesting finding of our study is the association found between the WW and the CW of cattle. We found that early weaners (100–150 kg) were associated with significantly lower CW. Further, they were kept longer in the grass to reach higher body weights. As a result of poor growth, cattle weaned at lower weights resulted in notably higher costs from post-weaning to slaughter. Even though early weaning is generally undertaken to optimize the reproductive performance of the mother [24], it can negatively impact profitability unless the growth of early weaners is carefully monitored to obtain higher weight gain during the post-weaning period. Sorting cattle by the weight category at weaning may assist farmers in better monitoring and maintaining animals’ weight gains, for instance, to decide targeted nutrition to meet specific weight gain goals. Such proactive actions may result in reduced costs and higher yields.

In the present study, two widely used methods—the MLR and the BRT—were considered to compare the forecasting outcomes. The BRT method is generally identified as a more robust method in forecasting, particularly with data consisting of complex non-linear predictor–response associations [13,18]. Nevertheless, this study did not find superior performance in the BRT over the MLR, which points to an absence of complex non-linear associations across the variables considered in the study. BRT models also provide flexible graphical tools to better understand predictor–response associations.

A major drawback identified in the study across the current management practices in the northern beef industry is the lack of regularly recorded body weight (BW) data at different stages of a cattle’s life. For instance, recording BW at least every six months starting from birth to slaughter would be highly beneficial for understanding cattle growth patterns, providing insights into improving the profitability of the beef supply chain. Studies [5,25,26] discussed that monitoring the growth and weight of cattle is significantly associated with the meat quality, profitability, and well-being of the cattle. The findings of our study have also shown the importance of weight data, particularly in forecasting, and highlight potential benefits for improved cattle monitoring and management practices. Nevertheless, current management systems face challenges in obtaining frequent BW measurements of cattle due to labor and time constraints. One potential solution for this problem involves employing previously discussed walk-over-weighing methods at regular intervals [27].

It is worth noting that certain factors such as birth weight [28], age at weaning [29], age at slaughter [30], records of diseases [31], and vaccination history [32] may potentially be associated with the growth performance of the cattle and thereby with the CW. Moreover, the quality of forage plays a pivotal role in determining the CW. Numerous studies have demonstrated a linear association between the nutritional content of forage and the live weight gain of beef cattle [33]. Nevertheless, these factors were not available to us in the study. Further, the current model with forecasts prior to one month of slaughter still has 16–18% of unexplained variation in the CW, which may be attributable to other characteristics; one key factor in this circumstance could be the weight loss of the cattle occurring closer to slaughter due to stress-related factors, such as changing environments and climates, loading to trucks, and separation from herds [34].

We used approximated weights for some cattle to assess the predictive performance before three months of slaughter; even though these values can be slightly different from actual weight values, effects arising from such errors were low to negligible. Further in the cost analysis, we only considered the grass and labor cost, but there are other small costs such as freight, repair, and maintenance that were not available for us in the study; nevertheless, these costs do not vary across cattle and apply to the entire herd.

The properties considered in the present study have mostly similar management practices related to grazing beef cattle, and hence no significant effects were observed in cattle across properties. While the impacts of predictive variables may vary depending on different practices, the methods and models presented in our study are widely applicable as previously discussed. In our future work, the models will be expanded for the southern beef industry in Australia; the southern industry differs from the northern beef industry related to different factors, such as management practices, diverse climate, and breed. In the present study, no impact from the weaning season was found likely due to the fact that many northern breeds are tropically adapted. Conversely, we expect significant seasonal impacts on the CW and other predictors from different breeds in the southern industry.

## 5. Conclusions

We showed that the carcass weight of grazing beef cattle can be predicted with better accuracy at different stages before slaughter by using readily available farm-level data. In summary, the findings of this study demonstrate that detailed cattle data can be better utilized to support decision-making in the beef supply chain with the use of statistical modelling, resulting in improved production outcomes. The statistical methodology developed in this study is directly applicable to grazing beef producers in Northern Australia. By offering actionable insights at the farm level, the adoption of this methodology is expected to enhance the productivity and profitability within the northern beef industry.

## Figures and Tables

**Figure 1 animals-13-01968-f001:**
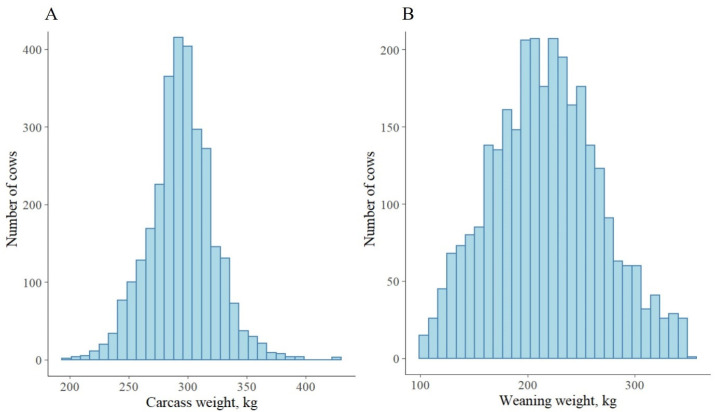
Distributions of the carcass weight (**A**) and the weaning weight (**B**).

**Figure 2 animals-13-01968-f002:**
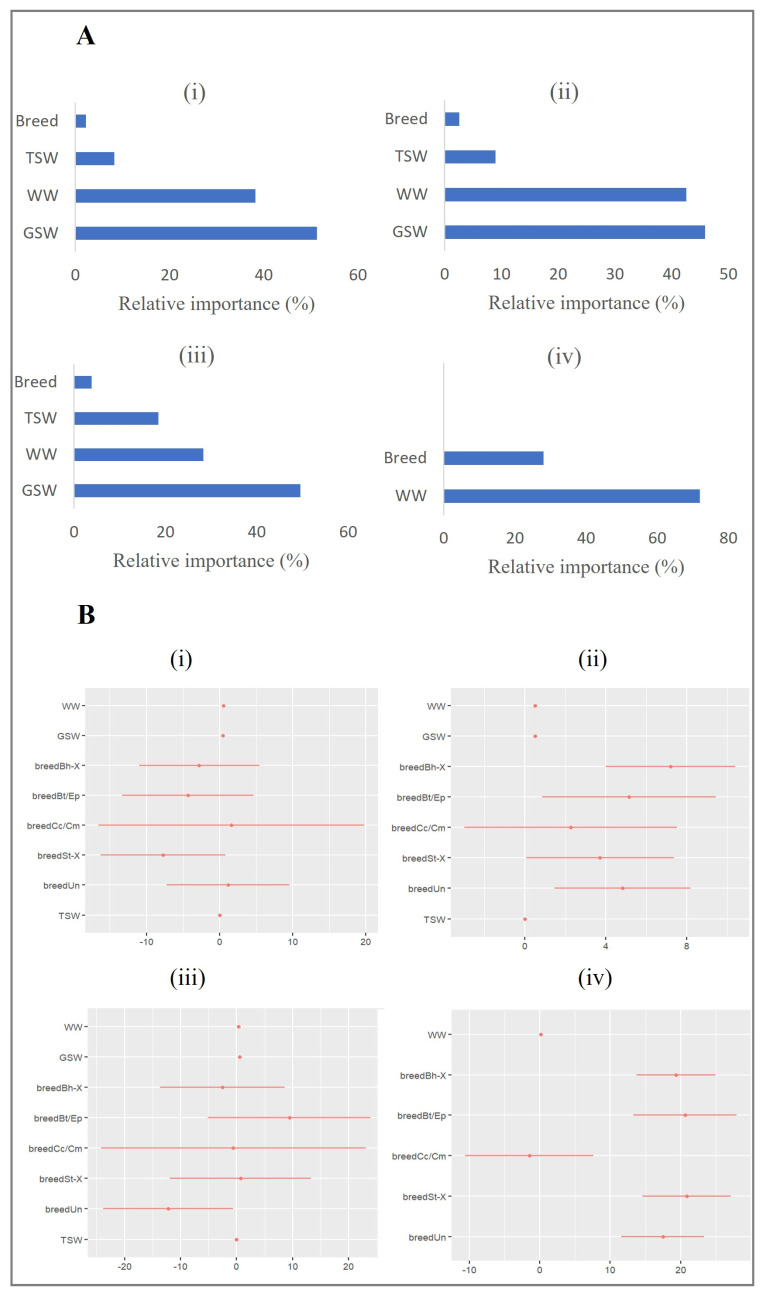
(**A**) Relative importance plots in the BRT model for 1 month before slaughter (**i**), 3 months before slaughter (**ii**), 9–10 months before slaughter (**iii**), and at weaning (**iv**). (**B**) Dot-and-Whisker plots of regression coefficients in the MLR model for 1 month before slaughter (**i**), 3 months before slaughter (**ii**), 9–10 months before slaughter (**iii**), and at weaning (**iv**).

**Figure 3 animals-13-01968-f003:**
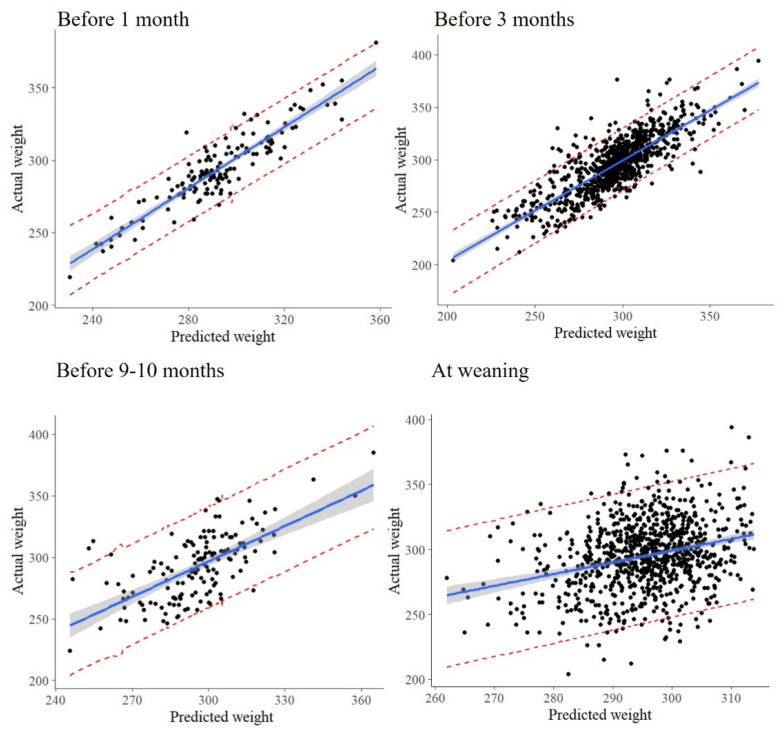
Actual versus predicted carcass weights with 95% level prediction intervals obtained using the MLR model.

**Figure 4 animals-13-01968-f004:**
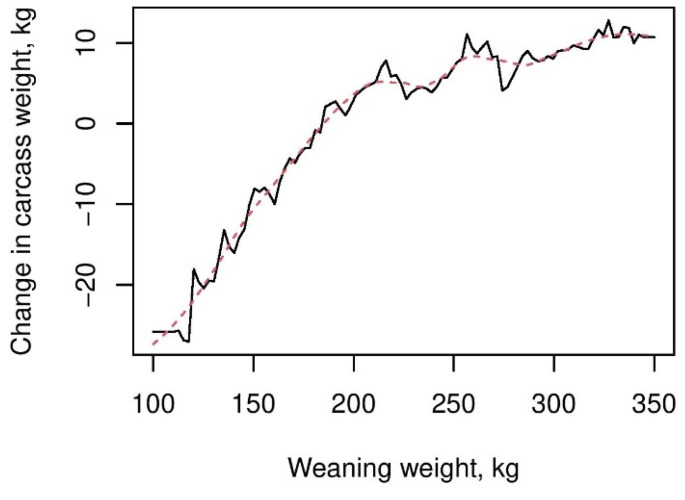
Association between the CW and the WW in scenario 4, obtained from BRT models.

**Table 1 animals-13-01968-t001:** Descriptive summary of the characteristics of the grass-fed beef cattle slaughtered between 2019 and 2021.

**Quantitative**				
**Feature**	**Min**	**Mean**	**Max**	**SD**
Carcass weight (kg)	197.0	295.0	426.0	27.1
Weaning weight (kg)	100.0	217.0	350.0	51.0
Pre_slaughter weight (kg)	400.0	554.0	802.0	46.8
Weight gain since weaning to slaughter (kg)	104.0	336.4	587.0	59.7
Average daily gain since weaning to slaughter (kg)	0.2	0.5	1.3	0.1
Time since weaning to slaughter (months)	10.2	25.0	47.1	5.8
**Qualitative**				
**Feature**	**P ^1^ (%)**	**Feature**		**P ^1^ (%)**
Breed		Weaned season		
Brahman (Bh)	4.5	Dry		61.8
Brahman cross (Bh-X)	48.6	Wet		38.2
British/European cross (Bt/Ep)	5.4			
Crossbreed/composite (Cc/Cm)	2.3	Property		
Santa cross (St-X)	12.7	A		73.3
Unknown (Un)	26.4	B		15.8
		C		10.9
Sex				
Female	2.7			
Male	97.3			

^1^ Proportion of the given category compared to the total number of cows in each feature.

**Table 2 animals-13-01968-t002:** Variance explained (R^2^) in carcass weight using the MLR and BRT models.

Scenario	Variance Explained in Carcass Weight—R^2^ (%)			
	MLR		BRT	
	BW Only	BW + Other Predictors	BW Only	BW + Other Predictors
Before 1 month of slaughter	77	82	78	84
Before 3 months of slaughter	70	74	71	80
Before 9–10 months of slaughter	30	46	39	59
At weaning	8	10	10	16

**Table 3 animals-13-01968-t003:** Predictive error of the carcass weight at different stages before slaughter.

Time before Slaughter	Sample Size	MLR			BRT		
		MAPE (%)	MAE (kg)	RMSE (kg)	MAPE (%)	MAE (kg)	RMSE (kg)
1 month (scenario 1)	483	2.67	7.90	10.48	2.98	8.72	11.65
3 months (scenario 2)	2995	4.12	12.04	15.57	4.03	11.78	15.34
9–10 months (scenario 3)	487	5.80	16.57	20.60	5.67	16.28	20.76
At weaning (scenario 4)	2995	7.79	19.67	25.29	7.68	19.38	25.06

**Table 4 animals-13-01968-t004:** Descriptive summary of costs (grass + labor) during the post-weaning-to-slaughter period by the weaning weight and weight gain categories.

Weaning Weight	Weight Gain: Weaning to Slaughter	Number of Cows	Carcass Weight per Head (kg)		Time on Grass per Head (Months)		Cost (Grass + Labor) per Head ^1^ ($)	
			Mean	SD	Mean	SD	Mean	SD
100:150 kg	200:300 kg	12	235	10.6	32.4	1.7	844.7	43.8
100:150 kg	300:400 kg	161	264	17.9	32.8	2.7	855.1	69.4
100:150 kg	400:500 kg	117	297	16.1	27.9	6.2	727.4	161.9
100:150 kg	500:600 kg	2	331	28.3	37.5	2.1	977.7	55.3
150:200 kg	200:300 kg	70	248	21.7	27.0	5.0	703.9	129.1
150:200 kg	300:400 kg	558	288	19.4	26.1	4.9	680.5	128.3
150:200 kg	400:500 kg	176	318	18.7	26.4	5.7	688.3	149.7
150:200 kg	500:600 kg	8	378	23.0	34.5	5.8	899.5	152.0
200:250 kg	100:200 kg	3	217	12.7	21.7	6.8	565.8	177.5
200:250 kg	200:300 kg	171	267	20.4	25.4	5.1	662.2	133.2
200:250 kg	300:400 kg	838	302	17.7	24.1	4.8	628.3	125.7
200:250 kg	400:500 kg	64	339	17.1	28.1	5.5	732.6	142.4
200:250 kg	500:600 kg	7	399	18.4	36.0	3.0	938.6	78.2
250:300 kg	100:200 kg	17	249	18.9	22.6	5.6	589.2	145.7
250:300 kg	200:300 kg	319	290	17.6	22.2	4.9	578.8	128.3
250:300 kg	300:400 kg	238	316	20.9	23.7	5.6	617.9	145.7
250:300 kg	400:500 kg	7	353	15.5	26.4	3.6	688.3	92.6
300:350 kg	100:200 kg	25	268	21.3	18.4	5.7	479.7	148.3
300:350 kg	200:300 kg	137	304	17.8	19.4	5.2	505.8	135.6
300:350 kg	300:400 kg	30	345	23.7	23.1	5.5	602.3	142.4

^1^ Cost cost (grass + labor) per month × time on grass; cost of grass + labor was considered as $6 per week; the costs values were obtained with the communication of end users.

## Data Availability

Restrictions apply to the availability of these data. Data were obtained from Black Box Co., Australia. To share the data, permissions from both the farmers and Black Box Co., are required.

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
