# Peer review of "Predicting Carcass Weight of Grass-Fed Beef Cattle before Slaughter Using Statistical Modelling"

_animals, 2023, doi:10.3390/ani13121968_

Round 1

Reviewer 1 Report

I have thoroughly read and critically assessed the manuscript on “Predicting carcass weight of grass-fed beef cattle before slaughter using statistical modelling”. An objective appraisal of this review indicates a well-written introduction that captures upfront, the main purpose of the study that clearly links with the topics in the manuscript. I find the introduction succinct, and it clearly teases out the problems in the industry that warrants attention. The idea of using statistical modelling to predict carcass weight is well thought off. The abstract summarises and links beautifully with the rest of paper in drawing the reader’s attention to the   relevance of the of the study. The logical organisation of the results that are easily flowing and easy to follow is quality. I find the discussion comprehensively argued with utilisation of discriminated previously published work in Spain and Brazil to compare and contrast their findings. The overall writing is clear and concise with correct grammar that flow throughout the paper. The only weakness is the lack of attention to minor details, Lines 4-5 authors name should not have titles. Italicise scientific names (Bos indicus). Where is reference 34? These minor details can easily reduce the quality of your work. The author needs to add more details in the conclusion, regarding the implication of the results of their present study. 

Author Response

Response to Reviewers’ Comments

We would like to express our sincere gratitude to the reviewer for the valuable comments, suggestions, and time devoted to reviewing our manuscript.

I have thoroughly read and critically assessed the manuscript on “Predicting carcass weight of grass-fed beef cattle before slaughter using statistical modelling”. An objective appraisal of this review indicates a well-written introduction that captures upfront, the main purpose of the study that clearly links with the topics in the manuscript. I find the introduction succinct, and it clearly teases out the problems in the industry that warrants attention. The idea of using statistical modelling to predict carcass weight is well thought off. The abstract summarises and links beautifully with the rest of paper in drawing the reader’s attention to the relevance of the of the study. The logical organisation of the results that are easily flowing and easy to follow is quality. I find the discussion comprehensively argued with utilisation of discriminated previously published work in Spain and Brazil to compare and contrast their findings. The overall writing is clear and concise with correct grammar that flow throughout the paper. The only weakness is the lack of attention to minor details,

  1. Lines 4-5 authors name should not have titles.

We thank the reviewer for pointing out this error. We have now removed the titles of the authors from the title page.

  1. Italicise scientific names (Bos indicus).

We have now converted these names to italic format. L116, and L309

  1. Where is reference 34? These minor details can easily reduce the quality of your work.

Thank you for the comment regarding the number of references in our manuscript. After carefully reviewing our originally submitted manuscript, we confirm that there were indeed only 33 references included in the submission. It appears that a typing mistake occurred during the subsequent stages of the submission. However, reference 34 in the present version is a result of the revisions made.

  1. The author needs to add more details in the conclusion, regarding the implication of the results of their present study. 

We appreciate the reviewer’s suggestion on this addition. We have revised the conclusion as suggested.

Reviewer 2 Report

The manuscript has scientific and practical relevance for beef industry, since predicting carcass weight anticipates decision-making on a farm. But, I think it can be improved. 

Some points to be considered:

In Introduction or material and methods is important writing  better and justify about regression methods, principally boosted regression trees. Is it a method non-linear? Why in conclusion you describe this method as machine-learning modelling? 

In modelling and forecasting should be improved, because it is difficult understand how subsets were divided. I think you were succinct, a detail about this division will be better, as well as the importance of different scenarios. 

Author Response

Response to Reviewers’ Comments

We would like to express our sincere gratitude to the reviewer for the valuable comments, suggestions, and time devoted to reviewing our manuscript. We have carefully addressed all the comments raised by the reviewer. Following are our systematic responses to the comments. Our responses were given immediately below the reviewers’ comments.

The manuscript has scientific and practical relevance for beef industry, since predicting carcass weight anticipates decision-making on a farm. But, I think it can be improved. Some points to be considered:

  1. In Introduction or material and methods is important writing better and justify about regression methods, principally boosted regression trees. Is it a method non-linear? Why in conclusion you describe this method as machine-learning modelling?

We thank the reviewer for this suggestion. We have revised the material and methods section to describe boosted regression tree method more broadly. L193-L202.

In the conclusion, we acknowledge that the use of the term "machine-learning" may not be appropriate since both models demonstrated similar accuracy, not solely the machine-learning method (boosted regression trees). L392-L393.

  1. Modelling and forecasting section should be improved because it is difficult to understand how subsets were divided. I think you were succinct, a detail about this division will be better, as well as the importance of different scenarios.

We thank the reviewer for suggesting this addition. We have now revised the modelling and forecasting section to provide a clear explanation of the division of the subsets and the importance of different forecasting scenarios. L135-L154.

Reviewer 3 Report

The statistical methodology presented in this manuscript may be novel to beef producers in northern Australia, but the results are far from novel. For example:

- Lower weaning weights result in lighter weight at slaughter, assuming grazing days are similar. Not novel.

- The closer cattle get to target slaughter weight (ex. 1 month vs. at weaning), the more accurately one can predict carcass weight. Not a novel thought.

A key unquantified variable contributing to model performance was consistency of daily nutrient intake (forage nutrient content during the grazing season(s) was not discussed). Apparently, nutrient intake (and concomitant daily gain) by cattle grazing the sub-tropical pastures of northern Australia is remarkably consistent, even though time on grass was as long as 3 years( Table 4). As the authors mention, when the predictive models are applied to cattle grazing a more variable environment (ex. southern Australia), I expect their utility to be limited. In more variable environments, forage quality and availability will have a significant impact on weight gain and the ability to predict harvest/carcass weight. 

All to say - I suspect practical application of the models and obtainment of meaningful results will be limited to the environment wherein the cattle were grazed. I concur with the authors - the models may be applicable to other sub-tropical grazing scenarios wherein daily nutrient intake is ~consistent over long (> 12 month) grazing periods.

Author Response

Response to Reviewers’ Comments

We would like to express our sincere gratitude to the reviewer for the valuable comments, suggestions, and time devoted to reviewing our manuscript. We have carefully addressed all the comments raised by the reviewer. Following are our systematic responses to the comments. Our responses were given immediately below the reviewers’ comments.

  1. The statistical methodology presented in this manuscript may be novel to beef producers in northern Australia, but the results are far from novel. For example: Lower weaning weights result in lighter weight at slaughter, assuming grazing days are similar - Not novel. The closer cattle get to target slaughter weight (ex. 1 month vs. at weaning), the more accurately one can predict carcass weight - Not a novel thought.

We appreciate and agree with the reviewer's points. The statistical methodology developed in our study brings a novel approach to the northern beef industry. We have revised the statement in the manuscript to explicitly convey this idea. We thank the reviewer for highlighting this fact. L305-L306.

  1. A key unquantified variable contributing to model performance was consistency of daily nutrient intake (forage nutrient content during the grazing season(s) was not discussed). Apparently, nutrient intake (and concomitant daily gain) by cattle grazing the sub-tropical pastures of northern Australia is remarkably consistent, even though time on grass was as long as 3 years (Table 4). As the authors mention, when the predictive models are applied to cattle grazing a more variable environment (ex. southern Australia), I expect their utility to be limited. In more variable environments, forage quality and availability will have a significant impact on weight gain and the ability to predict harvest/carcass weight. All to say - I suspect practical application of the models and obtainment of meaningful results will be limited to the environment wherein the cattle were grazed. I concur with the authors - the models may be applicable to other sub-tropical grazing scenarios wherein daily nutrient intake is ~consistent over long (> 12 month) grazing periods.

We thank the reviewer for drawing our attention to the influence of forage nutrient content on carcass weight. We have made additions to the discussion section of our manuscript to address this factor. L363-L366.